# Preserving Ready-to-Eat Meals Using Microwave Technologies for Future Space Programs

**DOI:** 10.3390/foods12061322

**Published:** 2023-03-20

**Authors:** Carolyn Ross, Shyam Sablani, Juming Tang

**Affiliations:** 1School of Food Science, Washington State University, Pullman, WA 99164, USA; 2Department of Biological Systems Engineering, Washington State University, Pullman, WA 99164, USA

**Keywords:** space foods, food safety, nutrition, thermal processing, sensory, packaging

## Abstract

The crewed suborbital and space flights launched by private companies over the past three years have rejuvenated public interest in space travel, including space tourism. Ready-to-eat meals (MREs) are the main source of nutrients and energy for space travelers. It is critical that those meals are free of bacterial and viral pathogens and have adequate shelf life. The participation of private companies in space programs will create new opportunities and demand for high-quality and microbiologically safe MREs for future space travels. In this article, we provide a brief review of nutrition and energy requirements for human activities in space. We discuss the general thermal processing requirements for control of bacterial and viral pathogens in MREs and introduce advanced thermal preservation technologies based on microwaves for production of MREs with different shelf-lives under various storage conditions. We also present the latest advancements in the development of polymer packaging materials for quality preservation of thermally stabilized MREs over extended storage. Finally, we recommend future research on issues related to the sensory quality of specially formulated MREs, microbial safety of dried foods that complement high moisture MREs, and food package waste management in future space missions.

## 1. Introduction

In the 1960s, both the United States and Soviet Union pioneered space technologies leading to human space missions. These early space missions lasted from a few hours to days, with nutrition to the astronauts provided in the form of pureed and squeeze-tube food formats [1]. Over the past few decades, humans spent several months to more than a year at the International Space Station (ISS) and other space systems. Those missions used canned and other shelf-stable foods for life support [2]. Since NASA’s future crewed missions to Mars require the prepared meals to have a minimum 5 year shelf life, the need for stable nutritious and appealing space-friendly meals becomes even more important [3].

In addition to the exploratory programs of NASA, several private companies have made significant investment in designing, developing, and deploying space systems for space flights. Since May 2020, SpaceX has transported 22 people to the ISS. In 2021, Blue Origin launched a vehicle into space with the capacity to transport four travelers. At the time this article was written in 2022, six companies are now arranging or planning to arrange civilian and tourist flights. The participation of private companies in space programs creates new opportunities in the development of high-quality and safe ready-to-eat meals for future space travels.

Prepared foods are often the only source of energy and nutrients supporting human activities in space missions, particularly in long duration missions. Yet, the microgravity, confined space, and limited storage and heating facilities influence the quality of eating experience and food intake. Since the first voyage into space, researchers have attempted to understand the influence of space conditions on human senses during space missions. To do this, studies have been conducted by astronauts via microgravity simulation [4]. Results across these studies were mixed for different reasons, including differences in methods of measurement, simulation of space conditions, and limited number of participants. As such, research is continuing in this area [5]. To address these continued, changing needs for space nutrition, this article presents a brief review of energy and nutrition requirements for astronauts, as well as potential novel thermal food preservation technologies and packaging solutions for future space missions.

## 2. Energy and Nutritional Requirements in Space Programs

The estimated energy requirement for astronauts on space missions requires several pieces of information [5]. Such information includes an activity factor of 1.25 (active), along with the astronaut’s age, body mass (kg), and height (m); resting metabolic rate, degree of movement and degree of daily exercise are also considered [6]. Considering this information, the diet supplied to astronauts during space flights is around 2800 kcal/day [5]. However, most space missions have fallen short of providing this goal number of calories [5,7]. Anecdotal reports of changes in appetite vary widely; Russian studies of Mir crewmembers showed that 40% reported no change in appetite, 20% reported increased appetite, and 40% of crewmembers reported decreased appetite [8]. This underconsumption of food during space flights is particularly problematic as this may result in the undernutrition of the astronaut and the risk of body mass loss, and more specifically, loss of lean mass and bone tissue [5]. The reasons for the calorie deficit are being explored but are not known at this time [9,10].

Considerations regarding nutrition for astronauts have developed over time. In the early days of the space program, developments in space nutrition were focused on weight and volume of the food. In the early space missions (Project Mercury, 1958–1963; Project Gemini, 1965–1966), the food supplied was exclusively dried food, with most products requiring water for rehydration [11]. As space programs developed and data regarding effects of spaceflight conditions on the human body became available, attention was increasingly paid to other aspects of food, including sensory properties and nutrition. This led to considerations regarding food preparation technology, shelf life, and packaging [12]. In the Apollo Space Program (1961–1972), menu items were expanded to include thermostabilized pouches, canned fruits, and irradiated meats [11].

Requirements for an astronaut’s diet may be considered as intrinsic properties of the food and the properties of the physical form of the food. From an intrinsic standpoint, the content of major essential nutrients should be balanced, have a high energy value, be palatable, and be easy to eat. Flavor and the factors affecting flavor perception in space have been extensively reviewed elsewhere [13]. While astronauts report that they sense a change in flavor during spaceflight, this can be difficult to define [14]. However, based on the earlier literature, foods with high flavor intensity are desired by astronauts [4]. Related to the packaging of food, factors to consider include minimal weight and volume, minimal waste, resistance to temperature and mechanical exposure, easy to prepare, with a long shelf life [12].

The Skylab (1973–1979) was the first U.S. space station to consider these requirements. Food quality was substantially improved over previous missions as the use of a freezer and refrigerator allowed the transport of frozen and refrigerated foods [11]. As a result, nutrient intake by astronauts was much higher than in previous missions.

The International Space Station menu presents meals with compositions that are acceptable for space flight. The ISS menu provides meals containing ~2800–3000 kcal/day. Of this, ~17% of total calories come from protein, 31% of total calories from fat, and 50% of calories from carbohydrates [11]. These meals were originally high in sodium (5300 mg/day). However, as research revealed that high sodium intake exacerbates bone loss, with possible pressure-induced vision changed, NASA reformulated foods to reduce sodium intake to 3000 mg/day. This change roughly aligns with the 2020–2025 Dietary Guidelines for Americans which recommends Americans consume less than 2300 mg/day (Centers for Disease Control).

Looking to future missions includes missions to Mars which are projected to last ~2.5 years, a combination of pre-packaged foods and some methods for growing foods will be required for a surface stay. As we look forward to these longer missions, research needs to continue in the establishment of nutritional standards and safe, nutritious, and palatable foods [11].

## 3. Thermal Processing Technologies for Shelf-Stable Ready-to-Eat Meals

Three preservation technologies, namely, freeze-drying, irradiation, and thermal processing (or retorting), have been used to produce single-serving shelf-stable ready-to eat meals (MREs) for astronauts on 4–11-month missions on the ISS [15,16]. Among those methods, thermal processing has produced meals with the highest acceptability and, therefore, holds the best potential in production of MREs for future long duration (e.g., 3–5 years) missions [3].

### 3.1. Conventional Thermal Processing, Principles, and Limitations

Thermal processing is a preservation method widely used in the food industry in production of shelf-stable high moisture content foods. The most heat resistant food-borne pathogens targeted in commercial thermal processes of low acid (pH > 4.6) and moist (water activity > 0.85) foods are *Clostridium botulinum* type A and B (proteolytic) spores [17] that may otherwise germinate and produce toxins in oxygen-free packages. The D_121C_ values (time required to cause one log10 reduction at 121 °C) for those spores are between 0.10–0.2 min [17], but D_121C_ = 0.25 min is often used in the calculations in developing commercial processes. In a thermal processing system, often referred to as a retort, prepackaged foods are exposed to pressurized steam or water typically at above 250 °F or 121.1 °C for a pre-determined amount of time selected by trained thermal processing specialists based on processing conditions, package geometry, and the food matrix. During this time, thermal energy is transferred from the package surface to increase the internal product temperature to a lethal level. The integrated thermal lethality at the cold spot in the food packages over the processing time is calculated as [17]:(1)$$F_{0}=\int_{0}^{t}10^{\frac{T\left(t\right)-121C}{10}}dt$$
where F_0_, in minutes, represents the accumulative time exposure of food at the cold spot to 121 °C; T(t), in °C, is the temperature measured at the cold spot in food packages as a function of processing time *t*, in minutes. Industrial thermal processes are designed to achieve F_0_ > 3 min, corresponding to a minimum of 12 log (=F_0_/D_121°C_) reduction in *C. botulinum* type A and B spores. To reduce spoilage likely caused by thermophiles, e.g., *Bacillus*, *stearothermophilus*, that are much more thermally resistant than *Clostridium botulinum* type A and B (proteolytic) spores, commercial processes are typically designed for F_0_ > 6 min in production of shelf-stable foods intended for long duration storage, particularly in warm environments. Such thermal processes are used in production of U.S. military MREs with an expected shelf life of 3 years.

Food matrices are, in general, relatively poor conductors of thermal energy, leading to slow heat transfer in food packages during thermal processing. For example, tuna packaged in 401 × 211 cans (103 mm diameter and 68 mm height) requires 100 min of heating in steam at 121 °C to become a shelf-stable product [18]. Thus, smaller food packages are more desirable in production of thermally stabilized foods.

Cook value C (in minutes) is used to evaluate the impact of a thermal process on food quality:(2)$$C=\int_{0}^{t}10^{\frac{T\left(t\right)-100C}{Zc}}dt$$
where z_c_ value, in °C, measures sensitivity of quality change rate influenced by temperature; it varies from 16 to 34 °C, depending upon the type of food and cook criteria (for example, taste, texture, or appearance) [19]. T(t) is the temperature history at a specific location in the food package selected for quality assessment. *C* values are often calculated by volumetric integration of quality losses throughout the whole volume of the containers, or for the most heated portion (i.e., adjacent to package) of the foods. The calculated value of *C* (mins) can be considered as an indicator of the quality change equivalent to that of the same food cooked at 100 °C for the same duration.

An example discussed in Tang (2015) [20] shows that 33 min is required to achieve F_0_ = 6 min for a food in 10 oz trays in a retort using 121 °C steam. The corresponding C values at the package surface is 212 min and at the package center is 93 min. Such a large difference in C values reflects the non-uniformity of the thermal exposure at different locations in the same package during thermal processing using retorts. In particular, the lengthy cooking near the package surface could cause severe thermal degradation in food quality. This may result in greater loss of heat sensitive nutrients and taste compounds.

Several pioneering food engineers have studied the various possibilities to minimize food quality losses, by varying processing temperature (for example, 113 to 145 °C) and package geometry [21,22,23,24,25]. Those studies concluded that compared to 121–125 °C typically used in commercial canning operations, high temperature retorting (e.g., 130–140 °C) could not significantly shorten processing times and result in better overall quality in solid foods in normal food package geometries because of the slow internal heat transfer within food packages. This leaves only two possible options to improve the quality of thermally sterilized foods: (1) using extremely thin profile food packages to facilitate heat transfer to the package center, or (2) changing heat transfer characteristics within food packages [21]. Single-serving pouches and shallow trays have since been used in commercial production of shelf-stable foods, in connection with the above first option. In particular, shelf-stable single-serving MREs for the U.S. military are exclusively thermally sterilized in flexible pouches. Although this advancement has significantly improved the quality of the MREs compared to canned foods, some of the quality attributes (e.g., texture and aroma) of the MREs require further improvements. The NASA Space Food Research Program has explored two alternative technologies to improve the quality of shelf-stable ready-to-eat meals, namely, pressure-assisted thermal sterilization (PATS) and microwave-assisted thermal sterilization (MATS) [26,27]. A comprehensive discussion on high pressure preservation of ready-to-eat meals can be found in a recent review article [28]. The following part of the article will only focus on microwave-assisted thermal processing technologies.

### 3.2. Microwave-Assisted Thermal Sterilization (MATS) Systems

Microwave heating is an efficient means of converting electric energy to thermal energy in food inside microwave transparent packages [20,29]. Novel in-package microwave processing technologies have been explored worldwide since 1970, specifically aiming at shortening thermal processing time and improving food quality. The historical development of different microwave processing systems for packaged foods have been reviewed in Tang (2015) [20]. The recent development of microwave-assisted thermal sterilization (MATS) systems based on 915 MHz single-mode microwave cavity design has led to FDA accepted processes for pre-packaged foods and a non-objection letter from USDA FSIS. Detailed information about the design of MATS systems and the procedures for process development and validation is provided in Tang (2015) [20]. The advantages of MATS were compared to conventional retorting in terms of shortened processing time and reduced cook values at both the cold and hot spots in 10 oz trays for two different processing temperatures (121 °C and 125 °C) when foods were processed in 10 oz (300 g) trays to F_0_ = 6 min [20]. The experimental data show that the cook values of the MATS-processed products were between 50 min at the cold spots and 58 min close to tray surface, much smaller than 78 min at the cold spot and 279 min close to packages in conventional retorting.

The improvements in the quality of two MATS-processed products (chicken breasts, chicken and dumplings) after a simulated 3-year storage were evaluated by professional sensory panelists at the U.S. Army Natick Soldier Center; the data are included in Tang (2015) [20]. The results showed superior quality of the MATS-processed foods compared to the counterparts by retort. But the data also indicate texture and flavor losses in the MATS-processed products over storage, highlighting a need to study and select appropriate food ingredients and packaging materials for stability of food qualities in MATS-processed meals during extended storage.

Due to the unique physiological conditions in space that impair human taste perception, astronauts on space missions may desire spicy meals with pleasing aroma [12]. Short-time MATS processing is particularly suited for producing shelf-stable spicy meals. Commercial MATS systems (Figure 1) are currently used in production of a wide range of authentic Indian meals under TATA Q brand (https://www.tataconsumer.com/brands/foods/tata-q, accessed on 7 March 2023). In addition to the improved taste profile, our recent studies have demonstrated that adding herbs allows for a 50% salt reduction in a MATS-processed chicken pasta meal while maintaining the same intensity of saltiness perception and overall meal acceptance [30].

### 3.3. Challenges in Developing Processing Schedules for MATS

Similar to the development of a conventional thermal process, measured temperature at the cold spot in packaged foods during microwave-assisted thermal processing is used in Equation (1) to calculate the processing time [19]. However, microwave heating presents two unique challenges that require special attention: (1) how to locate the cold spots in food packages, as the cold spots in foods are determined by microwave field distribution and not necessary in the geometric center of food packages; (2) how to accurately measure temperature at the cold spots in food packages in a continuous microwave system. To overcome the first challenge, we have developed a chemical marker method that measures brown color formation caused by Mallard reactions between reducing sugars and amino acids in model foods [31,32]. This method has been proven effective in determining cold spots for a wide range of food products [20]. We have also developed a mobile sensor method to accurately measure temperature at the cold spots in food packages for MATs [33,34]. Detailed procedures of process development and microbial validation for MATS are described in [20].

## 4. Pasteurized MREs for Space Programs

While shelf-stable meals for long duration exploratory space missions have been the main attention of the space food programs of National Aeronautics Space Administration (NASA) [3,15], the recent short duration space flights launched by private companies, such as Space X, Blue Origin, and Virgin Galactic, have brought attention to need for high quality MREs that do not require long shelf life. One of the main complaints about these private flights is a lack of good eating experiences. No cold storage and heating devices for foods were included in these pioneering flights launched by the private companies in 2021. These facilities may need to be considered in the design of future space vehicles. Leisure travelers are not professionally trained astronauts and may crave high-quality meals during flights in space. One of the most important considerations in MREs for these and other space flights is to minimize the risk of food poisoning [15]. Cold storage in combination with thermal pasteurization will provide opportunities for developing high-quality and microbiologically safe MREs for future space programs.

Processing conditions for thermal pasteurization of high moisture content read-to-eat meals are selected based on the target food-borne bacterial and viral pathogens and storage condition. *L. monocytogenes* is the most heat resistant vegetative bacterial pathogen that can survive freezing and grow in refrigerated conditions. The European Chilled Food Federation [35] recommends heating the product to 70 °C and holding at this temperature for 2 min as a minimum process condition that will cause 6 log reduction in *L. monocytogenes* in meals intended for 10 days storage above freezing but at or less than 5 °C. Hepatitis A virus (HAV) has been determined to be the most heat resistant viral pathogen in the published literature; heating a product to 80 °C and holding for 3 min would cause more than a 6-log reduction in HAV, Hepatitis E, and Norovirus [36]. Non-proteolytic *C. botulinum* spores are of a main concern in chilled meals stored for more than 10 days. ECFF (2006) [35] recommends heating foods to 90 °C and holding for 10 min to achieve a 6-log reduction in non-proteolytic *C. botulinum* spores in meals intended for storage of up to 6 weeks above freezing but at or less than 5 °C. The MREs pasteurized with the above processes can be stored frozen for extended periods without concern for microbial safety.

Prepackaged foods can be thermally processed using conventional in-package heating methods such as hot water flume or spray. Microwave-assisted pasteurization systems (MAPS) based on 915 MHz single-mode design are also developed at Washington State University (Figure 2) for short-time thermal processing to provide maximum quality retention.

A thermal process based on MAPS can raise the temperature of single-serving meals in 2–3 min to reach target pasteurization temperatures described above. A detailed design of MAPS and the procedures for process development are provided in Tang et al. (2018) [37]. Examples of meals thermally processed with MATS pilot-scale system are shown in Figure 3.

## 5. Packaging for Ready-to-Eat Meals in Space Programs

Currently, space foods are stored in ambient storage conditions (~23 °C and 1 atm). The expected shelf life of space foods is nine months to five years. Shuttle and ISS foods are required to have a minimum shelf life of nine months and one year, respectively. In contrast, extended duration missions to Mars will require a three to five year shelf life [38]. Both food packaging weight and waste are important issues for NASA and future space travels. Hence, low density packaging materials and packages that can be disposed of easily with small footprints are of great interest to future private and public space programs. But currently, metal and polymer containers in the form of cans, cups, bowls, lid films, and pouches are commonly used for the packaging of space foods.

### 5.1. Metal Packaging

Metal provides excellent barrier properties and can also withstand thermal sterilization conditions; hence, it is most used for thermostabilized foods. Commercial full-panel pullout aluminum cans and foil laminates are used for the Shuttle and ISS foods. Aluminum foil-based multilayer pouches are frequently used for MREs and popular among astronauts. Foil laminates with septum adapter have been used for beverages and rehydratable packages [38]. The septum facilitates the injection of water into the package during rehydration [38]. However, metals have higher density, which increases the package weight, and they also present challenges in the disposal of packaging waste.

### 5.2. Polymer Packaging

Polymeric packaging has attained higher industrial and consumer acceptability due to their oxygen and moisture barrier effectiveness, impact and flexural strengths, and transparency [39,40]. Compared to a metal fabrication process, the manufacturing of plastic containers can save a lot of production time as well as labor, energy, and costs. Polymers also provide more container design freedom than metals. Polymer packages are particularly suited for space missions because of their much lighter weights compared to metal and glass containers. Semi-rigid and flexible packaging also aids in trash compression. Polymeric packaging such as trays, cups, bowls, and pouches are frequently used for space foods including rehydratable and bite-sized foods, single-serving commercial condiments, and pudding containers.

Polymer packaging has the potential to entirely replace metal packaging for space foods. However, finite barrier properties and thermal resistance of polymeric packaging have been the limiting factors for its application for space foods. One of the key factors that affects the quality of packaged products in polymeric pouches, trays, bowls, and cups is the permeation of moisture and oxygen during storage. Moisture loss in packaged products during storage can reduce sensory food quality. Lengthy exposure to oxygen results in color changes such as browning; oxidation in foods with lipid content will cause rancidity, and nutritional losses [3,40,41,42,43,44]. In addition, packaging intended for thermostabilized foods must be able to withstand sterilization conditions.

Commercial polymer materials for food packages include polypropylene (PP), ethylene vinyl alcohol (EVOH), nylon 6, and polyethylene terephthalate (PET) [40,42,44]. As no single polymer intrinsically possesses the desired gas barrier and thermal and mechanical properties, these polymers are combined during the manufacturing stage to form multilayered structures [45]. Properly designed polymer-based packaging materials with superior barrier properties have been found to be suited for retort and advanced sterilization technologies such as MATS to produce high-quality MREs with extended shelf lives.

EVOH and metal oxide-coated PET films have been explored as as the oxygen barrier layer in polymer package materials; PP provides sealability and a water vapor barrier, and nylon imparts mechanical rigidity [46]. EVOH-based packages are made from a five-layer co-extrusion of nylon/ethylene vinyl alcohol/tie layer of polyethylene/linear low-density polyethylene. Single serving condiments, and rehydratable and bite-sized space foods using modified atmosphere techniques are packaged in EVOH packaging. Due to their moisture sensitivity, barrier properties of the EVOH-based films may deteriorate in thermal processing [44,46,47]. Hence, EVOH-based pouches are not ideal for in-packaged thermostabilized MREs intended for a shelf life of beyond one year. Our studies have found that the metal oxide-coated PET can provide a high barrier against oxygen and moisture and can retain the food quality for long duration [39,46]. Those packaging materials are transparent to microwaves and compatible with MATS and MAPS. Aluminum and silicon oxides are frequently used for developing the metal oxide-coated films. The metal oxide coating is deposited on the PET surface through chemical or physical vapor deposition or atomic layer deposition processes. Subsequently, the coated PET film layer is laminated with other film layers constituting of PP and nylon 6 to form the multilayered structure. Significant advances in improving the oxygen and water vapor barrier of polymeric films have been made in the past decade. Global polymer companies, including Amcor, DNP, Kuraray, Mitsubishi, Printpack, and Toppan, have significantly invested in research and development for metal oxide-coated PET-based high-barrier polymer packaging thermostabilized MREs. Some multilayer film structures have double and triple layers of coated PET, while other structures have incorporated oxygen-absorbing agents in the inside PP layer to improve oxygen barrier properties of packaging [39]. Polymer nanocomposites can provide alternative solutions in the field of high-barrier food packaging. Nanocomposites are obtained by dispersing inorganic–organic filler particles such as nanoclay in a polymer matrix such as PP, PET, EVOH, and nylons [48]. The U.S. Army Natick Soldier Research Center developed a five-layer coextruded film that incorporated 3.3–3.6% montmorillonite-clay to the MXD6 layer reducing the oxygen transmission rate (OTR) from 3.7 to 1.1 cm^3^ m^−2^ day^−1^ [49]. Nanocor developed a superior gas barrier structure, Imperm^®^ (Nanocor, Inc., Arlington Heights, IL, USA) by incorporating nanoclay in nylon 6, which has a four to five times superior oxygen barrier than that of nylon 6 [50].

The results from our recent research indicate that metal oxide-coated PET films can provide excellent oxygen barrier before and after sterilization processes, like that of the metal foil-based multilayer films. Many of these films maintained the U.S. Army specification of a 0.06 cc/m^2^/day oxygen transfer rate and a 0.01 g/m^2^/day water vapor transfer rate (WVTR) limit to have a shelf life of three years at 25 °C or six months at 37 °C (Table 1). These films facilitate commercial production of MREs using retort and microwave-based sterilization technologies.

### 5.3. Shelf Life of Thermostabilized MREs

Our recent studies assessed high-barrier packaging for military rations [37,40,44,54,55,56]. In collaboration with U.S. Army Natick Soldier Systems Center, scientists at Washington State University evaluated EVOH- and metal oxide-coated PET-based packaging in the retort and MATS processing of military rations. Several accelerated shelf life tests revealed that metal oxide-coated barrier packaging can provide a shelf life for ready-to-eat meals of at least 3 years, depending on the chemical composition of the recipe. Selected MREs were able to retain vitamins and sensory attributes for 3 years (Figure 4). Such high-barrier polymer packaging containing metal oxide is a suitable alternative to foil-based multilayer packaging for MREs [39,40,44,55,56]. However, more studies are needed to evaluate these advanced high-barrier packaging materials for MREs with five years of shelf life and other space foods for extended duration space missions to the Moon and Mars.

## 6. Needs for Future Research

To maintain the health of space travelers, it is imperative that adequate resources are provided to support food consumption on various space missions. A reliable food system must include the means to adequately prepare and store foods, as well as provide a variety of microbially safe yet palatable foods. Given the influence of sensory properties (i.e., how a food looks and tastes) on the acceptance of food, optimizing the sensory properties of a food will encourage consumption. There is a need for future studies on the influence of different food processing/methods and storage conditions of diverse food recipes to deliver safe and palatable meals to satisfy different needs of space travels. Spices can enhance the taste of MREs for space travels. However, the strong aroma from spicy meals may linger in the confined environment in space vehicles and stations and adversely affect the comfort of the space travelers. There is a need to develop and evaluate food recipes that take the advanced thermal processing technologies to enhance the taste of MREs while minimizing the effect on air quality. This issue may also be mitigated by adding aroma removal devices in the air circulation systems in space vehicles or stations.

Our above discussions have been focused on high-moisture foods. Low-moisture foods are important complements to high moisture ready-to-eat meals in space programs. For example, dried nuts, fruits, and other snacks have been used to complement prepared meals, along with freeze-dried foods, on previous space missions. Historically, low-moisture foods with water activity less than 0.6 were considered microbially safe. However, recent outbreaks and recalls caused by various pathogens, including *salmonella* and *cronobacter*, in low-moisture foods, e.g., chocolate, peanut butter, almonds, crackers, and baby formula, have created serious concerns about the safety of low-moisture foods and food ingredients. Certain bacterial pathogens, such as *Salmonella,* are extremely tolerant to heat in low-moisture foods [57,58,59,60,61,62]. *Salmonella*, in particular, can survive in a wide range of low-moisture foods, including walnuts, spices, and low-moisture military rations, during long time storage [63,64,65,66]. Future studies are needed to ensure the microbial safety of low-moisture foods for space programs.

Currently, astronauts manually collect food packaging and other waste materials and place it into bags and load it onto another vehicle designated for handling waste, which, depending on the flight, returns the waste to earth or burns it in the atmosphere. Long-duration space exploration missions will require innovative methods to recycle food packaging waste in space. Waste packaging is also converted for other purposes, including broken into water and oxygen and other gases which the crew can use or vent, as needed. In 2018, NASA sought new ideas to develop a high-temperature reactor that can function in microgravity and recycle waste [67]. Polymer packaging intended for space foods needs to be designed and developed so that it can easily be recycled in space and at the same time fulfill its shelf life requirements.

## Figures and Tables

**Figure 1 foods-12-01322-f001:**
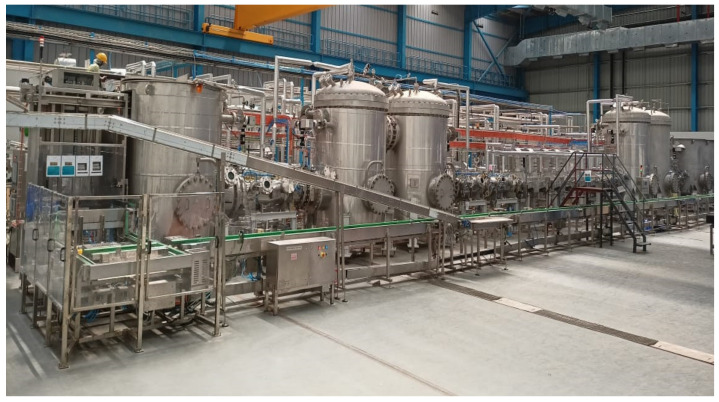
A continuous 915 MHz MATS system for production of shelf-stable meals (courtesy of 915 Lab).

**Figure 2 foods-12-01322-f002:**
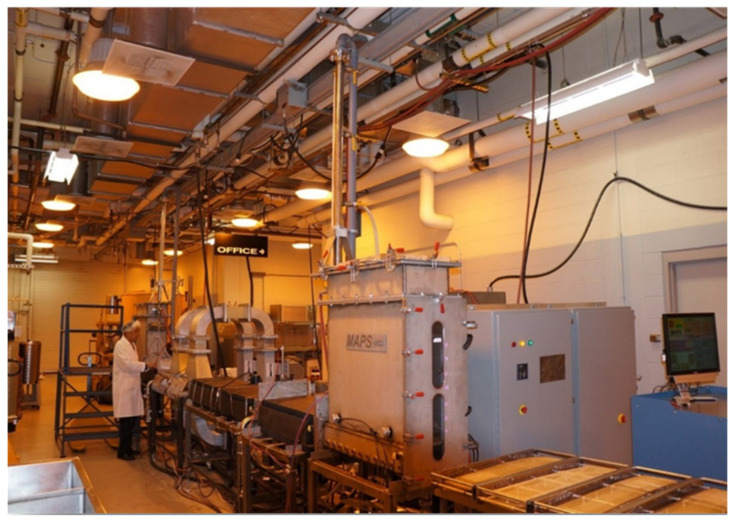
Pilot-scale 20 kW 915 MHz MAPs system at Washington State University.

**Figure 3 foods-12-01322-f003:**
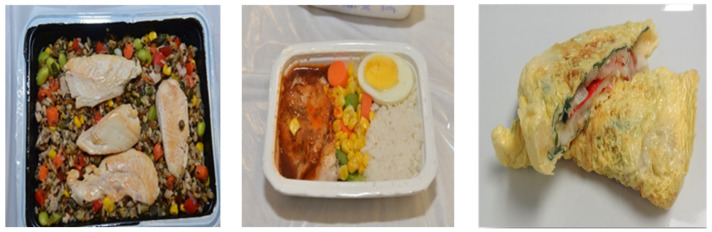
Thermally processed meals in 20 oz (herb chicken, pilaf, and vegetables—**left**) and 10 oz (egg, vegetables and rice—**center** and seafood omelet—**right**) trays using MAPS system.

**Figure 4 foods-12-01322-f004:**
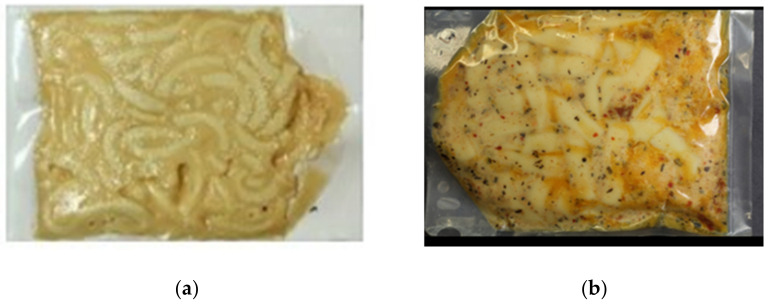
MATS-processed recipes in high-barrier polymeric pouches that could be stored for 3 years or more at room temperature conditions. These recipes in high-barrier packaging may be suitable for U.S. Army rations and NASA extended duration space missions (Reprinted/adapted with permission from Patel et al. [39,56]. (**a**) Mac and cheese in metal oxide-coated multilayer pouches (AlOx PET 12 μm/OPA 15 μm/CPP 70 μm); (**b**) Chicken pasta in metal oxide-coated multilayer pouches (AlOx PET 12 μm/AlOx PET 12/OPA 15 μm/CPP 70 μm).

**Table 1 foods-12-01322-t001:** Oxygen (OTR) and water vapor transmission rate (WVTR) of some high-barrier multilayer structures.

Main Composition	OTR cm^3^ m^−2^ day^−1^	WVTR, gm m^−2^ day^−1^	References
Double layer of metal oxide-coated films as barrier layers			
AlOx-coated PET/AlOx-coated PET/Oriented Nylon 6/CPP	<0.01	0.11 ± 0.02	[51]
AlOx-coated PET/AlOx-coated PET/AlOx-coated PET/Oriented-Nylon 6/CPP	<0.01	0.10 ± 0.003	[52]
Metal oxide coated and metallized film as a barrier layer			
Coated PET/Oriented-Nylon 6/CPP	0.04 ± 0.02	0.11 ± 0.01	[52]
SiOx-coated PET/Oriented-Nylon 6/CPP	0.02 ± 0.01	0.72 ± 0.06	[52]
Overlayer/AlOx-Organic-coated PET/Oriented-Nylon 6/CPP	0.02 ± 0.01	0.31 ± 0.02	[53]
Overlayer/SiOx-coated PET/Oriented-Nylon 6/CPP	0.01 ± 0.01	0.11 ± 0.01	[52]
Hyperbranched-PET/Oriented-Nylon 6/CPP	0.018 ± 0.001	0.440 ± 0.200	[43]
Composite coating PET/Composite coating Oriented-Nylon/PP	0.050 ± 0.007	5.190 ± 0.106	[43]
PET/Oriented-Nylon 6/PP	0.04 ± 0.01	0.38 ± 0.02	[47]
Coated-PET-Coated/Oriented-Nylon 6/PP	0.03 ± 0.01	4.15 ± 0.02	[47]
PET/SiOx coated PET/PP	0.18 ± 0.07	0.31 ± 0.06	[53]
PET-AlOx 12 µm/Oriented-Nylon 6/CPP	0.58 ± 0.11	0.16 ± 0.00	[53]
PET-AlOx/Oriented-Nylon 6/CPP	0.16 ± 0.05	0.08 ± 0.01	[53]
Metalized-PET/PE	1.33 ± 0.04	3.5 ± 0.01	[54]
EVOH as a barrier layer			
Oriented-Nylon 6/27% EVOH/CPP	1.120 ± 0.041	4.120 ± 0.092	[43]
PE/Oriented-Nylon 6/EVOH/Nylon 6/PE	0.16 ± 0.07	3.72 ± 0.06	[51]
PET/EVOH/PP	0.24 ± 0.03	0.73 ± 0.02	[47]
PET/PP/Oriented Nylon 6/EVOH/Oriented-Nylon 6/PP	0.11 ± 0.01	0.61 ± 0.01	[47]
Oriented-Nylon 6/EVOH/EVA	0.91 ± 0.10	4.51 ± 1.88	[54]

## Data Availability

No new data were created or analyzed in this study. Data sharing is not applicable to this article.

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
