# Peer review of "Preserving Ready-to-Eat Meals Using Microwave Technologies for Future Space Programs"

_foods, 2023, doi:10.3390/foods12061322_

Round 1

Reviewer 1 Report

In this research article (considered as a review article), the authors investigate the context of Ready-to-Eat-Meals dedicated to future space programs. The review article is structured with a general introduction presenting the future challenges that need to be addressed by the scientific community to meet the general requirements for future space programs in terms of food safety and quality. The following sections deal with well-knows scientific aspects concerning conventional retort processing and food safety. The section related to microwave processing (MATS and MAPS) is also well documented in previous papers from the same author (self-citations from the same author is also very used in the bibliography). The section related to packaging of MREs dedicated to future space programs only focus on the microwave technology as the only substitute to conventional retort processing. No other additional innovative technologies are mentioned, which is a lack for a review article.

In addition, the author do not discuss the effect of zero gravity on the storage of packaged foods as microgravity conditions encountered in space also affect the food packaging.

Author Response

This reviewer two comments:

Comment No. 1: only focused on microwave sterilization technologies. No other additional innovative technologies are mentioned, which is a lack for a review article.

Response: This is an invited review article specifically written for a special issue on microwave and ultra sounds. For clarification, we have changed the title of the review manuscript to "Preserving Ready-to-eat Meals Using Microwave Technologies for Future Space Programs".

We also have added " NASA Space Food Research Program has explored two alternative technologies to improve the quality of shelf-stable ready-to-eat meals, namely Pressure Assisted Thermal Sterilization (PATS) and Microwave Assisted Thermal Sterilization (MATS) [26,27]. A comprehensive discussion on high pressure preservation of ready-to-eat meals can be found in a recent review article [28]. The following part of the article will only focus on microwave assisted thermal processing technologies."  

Comment 2: In addition, the author do not discuss the effect of zero gravity on the storage of packaged foods as microgravity conditions encountered in space also affect the food packaging.

Response: Ready-to-eat meals are vacuum packaged. Zero gravity should make no different from storage on earth.

Reviewer 2 Report

This article is a relatively comprehensive and interesting review of the issue of ready-to-eat meals for astronauts. The impact of traditional pasteurization and sterilization processes on the nutritional and organoleptic as well as microbiological aspects of food are discussed. In addition, this article highlights the interest and advantages of microwave sterilization and pasteurization processes for ready-to-eat meals for astronauts from an organoleptic, microbiological, and processing time point of view. This technology makes it possible to achieve a better homogeneity of the heat treatment. This materializes by a temperature difference between the hottest and coldest point which is very significantly lower than that observed for sterilization in a retort. We therefore logically observe smaller differences in sterilizing and cooking values between these points.

Even if this article is clear and well presented, I suggest supplementing it with the following few elements:

It is known that in a conventional heating process, the cold point is generally located at the geometric center of the product. Thus, to guarantee the microbiological quality of the product, a minimum sterilizing value must be achieved for it, corresponding to a decimal reduction of 12 log for the spores of C. Botulinum. Thus, the control of sterilization in an autoclave is done by monitoring the temperature of the coldest point during sterilization using a probe positioned in the center of a tin can located in the center of the retort. Even if the equipment used for microwave sterilization has already been presented in Tang (2015), it would still be good if the authors discuss this point in the case of microwave sterilization and indicate the difficulties that can arise since the position of the coldest point is not easy to locate given that it varies according to the nature of the product and the shape of the container. Moreover, it would also be interesting to mention the problems that could arise with very heterogeneous products whose constituents have significantly different dielectric properties.

Line 147 - The reference (Lund 1986) has no associated number and does not appear to exist in the list of bibliographic references

Line 330 – I suggest changing “….layer reducing the OTR” to “…layer reducing the oxygen transfer rate (OTR)”

Line 336 – I suggest replacing “…water vapor transfer rate” with “…water vapor transfer rate (WVTR)”

Lines 337-338 – Say a word about the impact or not of the presence of metal oxide in the packaging on the homogeneity of the treatment

Author Response

We thank the reviewer for the positive comments.

The reviewer made a very good suggestion: "it would still be good if the authors discuss this point in the case of microwave sterilization and indicate the difficulties that can arise since the position of the coldest point is not easy to locate given that it varies according to the nature of the product and the shape of the container."

Response: we have added a new subsection:

3.3. Challenges in developing processing schedules for MATS

Similar to the development of a conventional thermal process, measured temperature at the cold spot in packaged foods during microwave assisted thermal processing is used in Equation 1 to calculate processing time [19]. But microwave heating presents two unique challenges that require special attention: 1) how to locate the cold spots in food packages, as the cold spot in foods are determined by microwave field distribution and not necessary in the geometric center of food packages; 2) how to accurately measure temperature at the cold spot in food packaged in microwave cavities in a continuous system. To overcome the first challenge, we have developed a chemical marker method that measures brown color formation caused by Mallard reactions between reducing sugars and amino acids in model foods [31,32]. This method has been proven effective in determining cold spots for a wide range of food products [20]. We have also developed a mobile sensor method to accurately measure temperature at the cold spots in food packages for MATs [33,34]. Detailed procedures of process development and microbial validation for MATS are described in [20]. "

We have also made corrections following the suggestions for line 147, 330 (of the original manuscript).

For suggestion related to line 337-338 in the original manuscript, we have added: "Those packaging materials are transparent to microwaves and compatible with MATS and MAPS."

Reviewer 3 Report

The manuscript presented for review with the title " Ready-to-eat Meals for future space programs" is very interesting and really important. A manuscript is a review type and has an extensive literature review.

One note is that the abbreviations used in Table 2 should be used in the title right next to the full names of these abbreviations.

Author Response

We thank the reviewer for the positive comment about the manuscript.

We have made change following the suggestion "One note is that the abbreviations used in Table 2 should be used in the title right next to the full names of these abbreviations"